

# SignVLM: a pre-trained large video model for sign language recognition

Hamzah Luqman

Information and Computer Science Department, King Fahd University of Petroleum and Minerals, SDAIA-KFUPM Joint Research Center for Artificial Intelligence, Saudi Arabia

## ABSTRACT

Sign language recognition (SLR) plays a vital role in including people with hearing impairment in the community. It facilitates the recognition of sign gestures and converts them into spoken languages. One of the main challenges for developing SLR systems is the lack of annotated datasets. This issue is more noticeable with low–resourced sign languages. To address this issue, we propose a pretrained large vision model, SignVLM, for SLR. This work explores the capability of the contrastive language–image pre-training (CLIP) model for SLR. This model is used to extract spatial features from the sign video frames while a Transformer decoder is used for temporal learning. The proposed model has been evaluated on four different sign languages using the KArSL, WLASL, LSA64, and AUSTL datasets. Different evaluation settings have been followed in this work including zero-shot and few-shot learning. The proposed model outperformed other models on the KArSL, WLASL, and LSA64 datasets and achieved comparable performance on the AUTSL dataset. The obtained results demonstrate the generalization of the proposed model to new datasets with few samples. The code and data are available at https://github.com/Hamzah-Luqman/signVLM.

## INTRODUCTION

More than 5.5% of the world population suffer from hearing loss (*World Health Organization, 2024*). Disabling hearing loss refers to hearing loss greater than 35 decibels (dB) in the better hearing ear. According to the World Health Organization, it is projected that one person in every 10 individuals will experience disabling hearing loss by 2050 (*World Health Organization, 2024*). People with hearing impairment often face challenges in communication, education, and social participation, leading to isolation and lower quality of life.

Sign language plays a vital role in facilitating communication for individuals with hearing loss, particularly those who are deaf (*El-Alfy & Luqman, 2022*). This language depends mainly on hand gestures and body language. Facial expressions are also used simultaneously with other gestures to convey some grammatical meanings that couldn't be expressed by hand, such as emotion and negotiation. Sign language is a complete language with vocabulary, structure, and grammar that differ from spoken languages. The

Corresponding author
Hamzah Luqman,
hluqman@kfupm.edu.sa

equivalent of a natural language word in the sign language is called *gloss*. Sign languages do not have a strong relationship with spoken languages. This justifies having different sign languages for countries that share the same spoken languages, such as British and American sign languages. Sign language is more related to the environment and culture of the country, which forms some sign gestures. Therefore, we can find some variations between sign languages within the same country, similar to the dialectical variations between spoken languages.

Integrating deaf people or people with hard hearing loss into society requires a bi-directional recognition and production system. The sign language recognition (SLR) system recognizes sign gestures and converts them into their equivalent spoken words. The sign language production system translates the spoken language, text or speech, into a sign language. The avatar is used usually to visualize the signs generated by the production system. SLR involves recognizing sign gestures in the video stream. This task is more challenging than production due to the difficulties inherited from the video understanding problem. However, other issues make sign language more challenging than action recognition and video understanding. These challenges involve variations between signers and signs performed by different signers, changes in the sign gesture based on its context, and the use of non-manual gestures. However, the lack of an annotated dataset is the most challenging issue for developing SLR systems.

To address the lack of annotated datasets, several SLR approaches have recently been proposed utilizing semi-supervised learning techniques (*Bird, Ekárt & Faria, 2020*; *Selvaraj et al., 2022*; *Bilge, Cinbis & Ikizler-Cinbis, 2022*). These approaches leveraged the large available unannotated data to boost the performance of the recognition system. *Bilge, Cinbis & Ikizler-Cinbis (2022)* proposed an embedding-based framework for zero-shot SLR. This approach uses two separate streams for visual and textual sign representation. The visual component uses spatiotemporal models to encode body and hand regions, while the textual embeddings are obtained by encoding the textual dictionary definitions and attributes of each sign. However, this approach requires a textual description of each sign in the dictionary which is challenging for sign languages with large vocabulary. In addition, the performance of such techniques does not compete with the full-supervised based learning techniques.

Large vision models, such as contrastive language–image pre-training (CLIP) model (*Radford et al., 2021*), have made substantial advancements in various computer vision tasks (*Zhao et al., 2023b*; *Shen et al., 2021*). By learning to associate images with natural language descriptions, CLIP builds a shared understanding of both visual and textual information. This multi-modal learning approach enables CLIP to excel in a range of tasks, from image classification (*Abdelfattah et al., 2023*) and object detection (*Wu et al., 2023*) to more complex applications like image captioning and visual question-answering (*Shen et al., 2021*). To leverage the powerful multi-modal learning capabilities of CLIP, we propose SignVLM, a pre-trained large video model for SLR. The proposed model utilizes the CLIP model for extracting spatial features from the sign video frames while the Transformer decoder, inspired by *Lin et al. (2022)*, is used for temporal learning. The proposed model has been evaluated on four datasets representing different sign languages.

Different evaluation settings have been followed in this work starting from zero-shot learning to a full fine-tuning of FrozenCLIP. The proposed approach outperformed state-of-the-art (SOTA) models on three out of four datasets. This shows the advantage of utilizing large vision models for limited data sign languages. Our main contributions can be summarized as follows:

- Proposing SignVLM, a pre-trained vision model for SLR using limited labeled data
- Leveraging CLIP for spatial feature extraction and a Transformer decoder for temporal modeling
- Evaluating the model on four datasets with different evaluation settings
- Outperforming SOTA models on three out of four datasets

This article is organized as follows. The related work is presented in 'Literature Review' and the SLR problem is defined in 'Sign Language Recognition'. 'Proposed Approach' details the proposed model and the experimental results are presented in 'Experimental Work'. The conclusions are presented in 'Conclusions'.

## LITERATURE REVIEW

The emergence of advanced deep learning frameworks has led researchers to apply convolutional neural networks (CNNs) followed by recurrent neural networks (RNNs) for SLR. CNNs are used to extract spatial features from video frames (*Arooj et al., 2024*; *Jia & Li, 2024*; *Luqman & El-Alfy, 2021*; *Rastgoo, Kiani & Escalera, 2020a*; *Zhang & Zhang, 2021*), while RNNs are employed for temporal learning (*Luqman & El-Alfy, 2021*; *Rastgoo, Kiani & Escalera, 2020b*; *Aly & Aly, 2020*; *Suliman et al., 2021*; *Lee et al., 2021*; *Chaikaew, Somkuan & Yuyen, 2021*). Some studies have utilized 3D-CNNs to capture both spatial and temporal information simultaneously (*Huang et al., 2019*; *Rastgoo, Kiani & Escalera, 2020a*; *Jiang et al., 2021*; *Zhang & Zhang, 2021*). The Inflated 3D ConvNet (I3D) model (*Carreira & Zisserman, 2017*), initially designed for action recognition, has been adapted for various SLR applications (*Li et al., 2020a*; *Joze & Koller, 2020*; *Adaloglou et al., 2021*). A comparative study (*Li et al., 2020a*) has shown that I3D outperforms graph convolution network (GCN) based models in SLR. *Özdemir, Baytaş & Akarun (2023)* introduced a pose-based GCN model that integrates GCN with multi-cue long short-term memory networks (MC-LSTMs) to process cues from the hands, body, and face, achieving 90.85% accuracy on the AUTSL dataset with 100 classes.

The success of Transformer networks across various fields has led researchers to explore Transformer-based frameworks for SLR (*Shin et al., 2023*; *Boháček & Hrúz, 2022*; *Camgöz et al., 2020*; *de Coster, van Herreweghe & Dambre, 2020*; *Aloysius, Geetha & Nedungadi, 2021*; *Selvaraj et al., 2022*; *Zhou, Tam & Lam, 2021*; *Tunga, Nuthalapati & Wachs, 2021a*). *de Coster, van Herreweghe & Dambre (2020)* compared the performance of RNNs, Transformers, Pose-based Transformers, and multi-modal Transformer networks. Their study showed that the multi-modal framework, which utilized pose information and features extracted through 2D CNNs, achieved the highest performance with an accuracy of 74.70%. *Tunga, Nuthalapati & Wachs (2021a)* integrated a GCN with a BERT-based

model to encode signer pose data, resulting in a 60.1% accuracy on the Word-Level American Sign Language (WLASL) dataset.

Although fully supervised SLR models have demonstrated high accuracy, their dependency on large annotated datasets restricts their applicability to many sign languages. According to *El-Alfy & Luqman (2022)*, most of the SLR studies targeted American (*Li et al., 2020b*) and Chinese (*Rastgoo, Kiani & Escalera, 2021*) sign languages. This can be attributed to the availability of large annotated datasets for these languages. However, this is not the case for most sign languages where resources are scarce. The collection and annotation of SLR datasets is an expensive and time-consuming task due to the lack of sign language experts who can annotate datasets at the sign or gloss levels.

To tackle the issue of signer dependence and environment constraints that are captured by RGB data, researchers started to use pose data for SLR. Recently, several pose-based SLR methods have been proposed (*Tunga, Nuthalapati & Wachs, 2021b*; *Selvaraj et al., 2022*; *Hu et al., 2021*; *Zhao et al., 2023a*; *Jiang, Camgoz & Bowden, 2021*). *Tunga, Nuthalapati & Wachs (2021b)* used GCN to model the sign gesture represented as a sequence of pose data. *Kindiroglu et al. (2024)* used GCN with the pose data for knowledge sharing between different datasets. *Laines et al. (2023)* converted the pose data into an RGB image and utilized an RGB-based approach to model the sequential pose data. SignFormer (*Kothadiya et al., 2023*) used the transformer encoder for recognizing static Indian sign language. *Alyami & Luqman (2025)* proposed a Swin-MSTP formwork for SLR. The proposed method uses a Swin Transformer for spatial feature extraction and a multiscale temporal perception for temporal learning. BEST (*Zhao et al., 2023a*) utilized BERT for SLR with pose data. SignBERT (*Hu et al., 2021*) and SignBERT+ (*Hu et al., 2023*) models have been proposed for SLR. These models masked some pose data and trained the model to reconstruct the masked pose data.

*Abdullahi et al. (2024)* used feature selection with end-to-end Fourier CNN to mitigate the redundant spatial-temporal feature pruning problem that results in missing the semantic and temporal dependencies. This problem usually causes signs misclassification. The spatial-temporal contextual information was captured by *Miah et al. (2024)* using a two-stream multistage graph convolution with attention and residual connection. Graph convolution network with pose data has also been used by *Naz et al. (2023)*. *Zuo, Wei & Mak (2023)* developed an SLR system that exploits semantic information of sign glosses. The proposed approach generates soft labels for signs with similar glosses. Then, an inter-modality mixup technique was used to blend vision and gloss features and recognize each sign.

Low-data learning methods offer a solution to the challenge of limited data availability in certain sign languages. These methods train models with a small amount of labeled data by employing techniques like transfer learning and data augmentation to improve performance. WiSign (*Shang & Wu, 2017*) utilized a co-training semi-supervised learning (SSL) approach with support vector machines (SVM) and k-nearest neighbors (KNN). In this approach, the classifiers are trained on labeled data and then used to predict labels for unlabeled instances. When both classifiers agree on a label for an instance, that instance is

labeled accordingly and added to the labeled dataset. This iterative process continues until all instances are labeled.

*Selvaraj et al. (2022)* investigated the use of transfer learning from Indian sign language to other languages, employing self-supervised pretraining strategies. Their findings showed that cross-lingual transfer learning significantly improved accuracy in low-resource languages, with gains ranging from 2% to 18%. The authors also introduced OpenHands, an SLR library featuring various pre-trained models based on pose data, including LSTM, Transformer, and two graph-based architectures. Another study (*Bird, Ekárt & Faria, 2020*) also explored cross-lingual transfer learning, presenting a late fusion model for British SLR. This model combined RGB input processed by VGG16-MLP with Leap data modeled by an evolutionarily optimized deep multi-layer perceptron (MLP), using late fusion to predict the final sign class. This approach demonstrated the effectiveness of transfer learning from British to American sign languages, achieving 94.44% accuracy on a small dataset of 18 signs.

A recent study (*Bilge, Cinbis & Ikizler-Cinbis, 2022*) examined zero-shot learning for SLR, where the model is trained to recognize signs not present in the training data. The framework used four data streams: two for visual descriptions from sign videos and two for textual descriptions from the Webster sign language and American Sign Language hand shape dictionaries. Various models were tested for spatio-temporal video modeling, and a BERT architecture was used for textual modeling. While the results indicated promising potential for zero-shot sign recognition, the accuracy achieved was relatively modest compared to other domains utilizing zero-shot learning. *Zheng et al. (2023)* proposed a contrastive visual-textual transformation for SLR. The proposed model utilizes the pre-trained knowledge of visual and language modalities to boost recognition accuracy.

Table 1 summarizes the surveyed studies. Despite these advances, current SLR research faces several challenges. One major limitation is the lack of large annotated datasets, especially for low-resource sign languages. Most existing studies focus on well-resourced languages such as American and Chinese Sign Language, leaving other languages underrepresented. While Transformer-based and pose-based approaches have been extensively explored, the use of large pre-trained vision-language models like CLIP remains in its early stages for isolated SLR. These models hold significant potential for addressing the issue of data scarcity in SLR. However, current zero-shot learning methods for recognizing unseen signs yield modest results and require better integration of pre-trained vision-language models. Furthermore, high computational demands limit the real-time deployment of many models. There is also a lack of standardized benchmarks that reflect diverse linguistic and environmental conditions. Addressing these gaps is essential for developing more data-efficient, generalizable, and real-time SLR systems that are accessible across a broader range of use cases.

## SIGN LANGUAGE RECOGNITION

SLR is a computer vision problem that aims to automatically interpret and translate sign language into written or spoken language. SLR systems can be categorized into two main types: isolated sign recognition, where individual signs are identified, and continuous SLR,

**Table 1 Summary of the surveyed studies.**

| Study | Sign language | Input modality | Data efficiency | Dataset | Number of signs | Accuracy (%) |
|---|---|---|---|---|---|---|
| *Bilge, Cinbis & Ikizler-Cinbis (2022)* | Generic SLR | RGB + Text | ✓ | – | 263 | 73.4 |
| *Hu et al. (2021)* | ASL | Pose | ✗ | MSASL | 1,000 | 57.1 |
| *Zhao et al. (2023a)* | ASL | Pose | ✗ | MS-ASL | 1,000 | 77.9 |
| *Li et al. (2020b)* | ASL | Pose | ✗ | WLASL | 100 | 55.4 |
| *Naz et al. (2023)* | ASL | Pose | ✗ | WLASL | 100 | 72.1 |
| *Kothadiya et al. (2023)* | Indian | Pose | ✗ | – | 36 | 99.3 |
| *Selvaraj et al. (2022)* | Multiple | Pose | ✓ | – | – | 91.2 |
| *Sidig et al. (2021)* | Arabic | RGB | ✗ | KArSL | 190 | 25.6 |
| *Luqman (2022)* | Arabic | RGB | ✗ | KArSL | 502 | 32.7 |
| *Alamri et al. (2024)* | Arabic | RGB | ✗ | KArSL | 502 | 30.5 |
| *Li et al. (2017)* | Generic | RGB + Depth | ✗ | – | – | 98.8 |
| *Marais et al. (2022)* | Argentinian | Pose | ✓ | LSA64 | 64 | 81.2 |
| *Shah (2018)* | Argentinian | RGB | ✗ | LSA64 | 14 | 96.0 |
| *Akdag & Baykan (2024)* | Argentinian | Pose | ✗ | LSA64 | 64 | 98.9 |
| *Sincan & Keles (2020)* | Turkish | RGB | ✗ | AUTSL | 226 | 47.6 |
| *Vázquez-Enríquez et al. (2021)* | Turkish | RGB | ✗ | AUTSL | 226 | 90.3 |
| *Sincan & Keles (2022)* | Turkish | RGB | ✗ | AUTSL | 226 | 93.5 |

where full sentences and phrases are translated, taking into account the temporal dependencies and context of each sign. In this work, we will target isolated SLR.

SLR accepts an input video containing a sign gesture and outputs the meaning of the sign. *Gloss* is used to refer to the meaning of the sign gesture in the spoken language. Isolated SLR is a many-to-one problem, where a set of frames $X = \{x_1, x_2, x_n\}$ are mapped to one sign word or gloss $Y = \{y\}$. Recognizing sign gestures requires spatial and temporal learning. Each sign gesture can consist of manual and non-manual components. The manual component involves hands' motion while the non-manual component involves facial expressions and body movements. Learning both components requires an efficient features extractor in the spatial and temporal spaces.

## PROPOSED APPROACH

The proposed SignVLM consists of two main components, frame encoding and temporal learning, as shown in Fig. 1. This architecture combines the strengths of CLIP's pre-trained visual representation with advanced temporal modeling to handle the intricate dependencies in sign language gestures, making it well-suited for accurate and robust SLR.

**CLIP visual encoder.** Each frame of the sign video is fed into the CLIP model (*Radford et al., 2021*) to extract spatial features. CLIP is a large-scale vision-language model trained on 400 million image-text pairs collected from the internet. The key innovation of CLIP lies in its contrastive learning approach, which enables it to learn

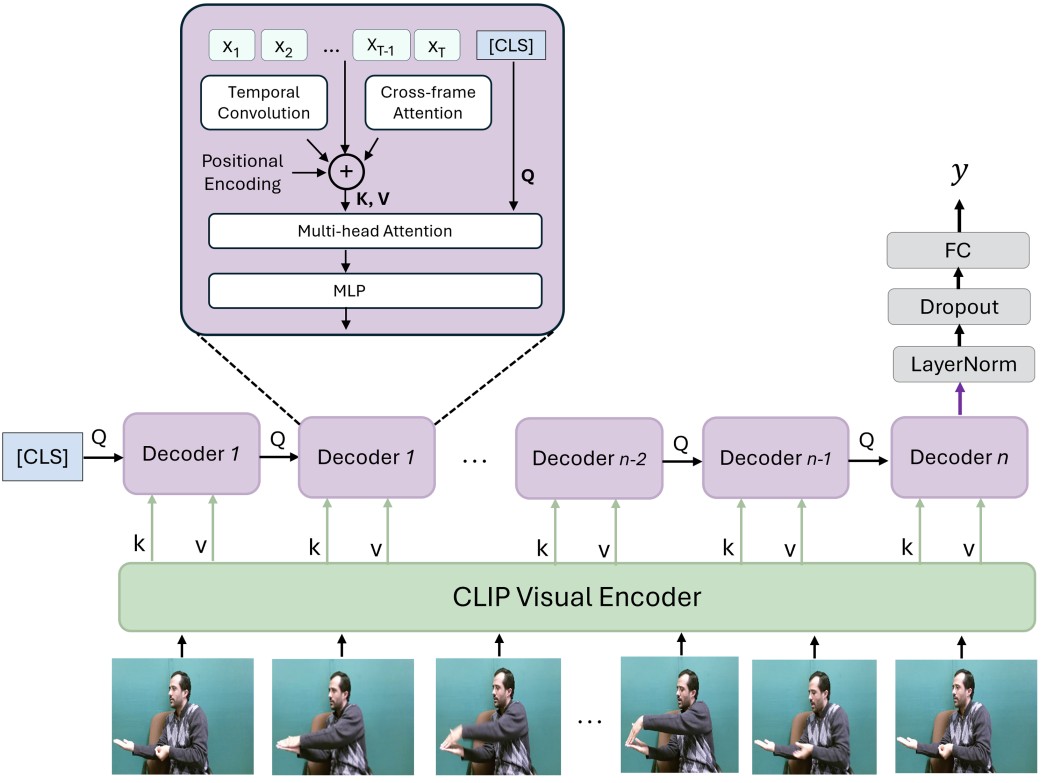

**Figure 1 The framework of SignVLM.** The CLIP encoder extracts visual features from each frame in the sign video. These features are fed into transformer decoders to learn the temporal information of the sign. Adapted from *Lin et al. (2022)*.

generalized visual representations by associating images with their corresponding text descriptions.

CLIP consists of two main components: a vision encoder, which is a vision transformer (ViT) backbone that processes input images to extract visual features, and a text encoder, which is a causal language model that processes input text (*e.g.*, captions or sign glosses) to extract textual embeddings. Both encoders project their respective outputs into a shared latent space of identical dimensions. The core idea behind CLIP's training is that image and text embeddings that correspond to the same concept should be close in this shared space, while unrelated embeddings should be far apart. The dot product of these embeddings is used as a similarity score to align correct image-text pairs. CLIP's learning process does not rely on task-specific supervision, making it highly effective in zero-shot settings—where it can generalize to unseen tasks without requiring additional training data. This is particularly beneficial for SLR, where annotated datasets are often limited.

One of the key advantages of using CLIP is its ability to learn generalized feature representations. Unlike traditional vision models trained on task-specific datasets, CLIP learns rich, transferable representations from diverse image-text pairs, which helps in SLR,

where visual cues are complex and often lack large-scale labeled data. Furthermore, CLIP is inherently designed for zero-shot learning, meaning it can generalize well to unseen data without extensive fine-tuning, which is crucial for recognizing new or underrepresented signs in datasets.

Since CLIP learns by differentiating between correct and incorrect image-text pairs, it develops robust feature representations that help reduce sign ambiguity and improve recognition accuracy. The image encoder of the pre-trained CLIP model is leveraged in our framework to extract spatial features from each sign video frame. These extracted features serve as input to the temporal learning module, which models sequential dependencies between frames, leading to improved sign recognition accuracy. Another significant advantage of using CLIP is its ability to eliminate the need for extensive task-specific labeling. Traditional SLR models require manually labeled datasets, which can be time-consuming and expensive to create. By leveraging pre-trained CLIP embeddings, our approach reduces dependency on extensive labeled sign language datasets, making it more scalable and efficient.

**Temporal learning decoder.** The extracted vision features using the CLIP encoder are fed into the temporal learning decoder, which is designed to capture temporal dependencies across the sequence of frames. Inspired by *Lin et al. (2022)*, this decoder consists of temporal convolution, cross-frame attention, multi-head attention, and an MLP layer.

Temporal convolution layers capture short-range dependencies by processing a local window of frames, allowing the model to recognize short temporal patterns in gestures. The cross-frame attention layer attends to features across the entire frame sequence, enabling the model to understand long-range dependencies and context across the video. This attention mechanism, applied across frames, allows for effective aggregation of frame-wise features. Positional encodings are added to the input frame features to retain the order of frames in the sequence. These encodings are important for temporal understanding in sequence-based tasks. The multi-head attention and MLP layers help capture complex interactions and patterns in the encoded feature sequence. Multi-head attention enables the model to focus on multiple aspects of the frame sequence simultaneously, while the MLP further refines these features.

**Classification.** A special classification token (Classification (CLS)) is introduced, which learns to aggregate the information across all frames for final classification. The processed sequence, along with the classification token, is fed through multiple decoders of the temporal decoding component. Each decoder layer contributes to refining the sequence representation and enhancing the temporal understanding, with the final decoder output used for classification. The output from the final decoder is passed through a LayerNorm layer, followed by a dropout technique for regularization, and then a Fully Connected (FC) layer for classification. This final FC layer produces the probability distribution or predicted class for the sign language sequence that corresponds to the recognized sign.

# EXPERIMENTAL WORK

**Datasets.** To evaluate the effectiveness and generalization of the proposed sin language recognition approach, we evaluated it on four prominent sign language datasets: KArSL (https://hamzah-luqman.github.io/KArSL/) (*Sidig et al., 2021*), WLASL (https://dxli94.github.io/WLASL/) (*Li et al., 2020b*), LSA64 (https://facundoq.github.io/datasets/lsa64/) (*Ronchetti et al., 2016a*), and AUTSL (https://cvml.ankara.edu.tr/datasets/) (*Sincan & Keles, 2020*) datasets. Each dataset represents a distinct sign language and varies in vocabulary size, signer diversity, and recording conditions, which offer a comprehensive benchmark for our approach.

KArSL is an Arabic sign language dataset. It consists of 502 signs performed by three signers. Each sign was repeated several times in different sessions to add more variations to the dataset's signs. The dataset was collected using Kinect V2 and it is available in three modalities: RGB, depth, and skeleton information. Three versions of the dataset are available with a different number of signs: KArSL-100, KArSL-190, and KArSL-502. We evaluated our approach on the three sets. We also used the WLASL-100 dataset which is one of the most widely used datasets for ASL recognition. The dataset consists of 100 signs performed by several signers in various dialects and signing styles within ASL. The LSA64 dataset is an Argentine sign language dataset. It consists of 3,200 videos representing 64 signs, derived from 10 non-expert signers. We evaluated our approach on each signer in the signer-independent evaluation setting and the average accuracy of all 10 signers is reported. AUTSL dataset is a multimodal Turkish dataset consisting of 226 sign gestures performed by 43 signers under different conditions with different backgrounds. Each signer repeated each sign 169 times on average which resulted in 36,302 samples. The dataset was split into train and test with 28,142 and 3,742 samples, respectively. The dataset is available in three modalities, RGB, depth, and skeleton information. In this work, we used the RGB modality. A summary of these datasets is available in Table 2. As shown in the table, the used datasets consist of different sign languages with a variation in the number of signs, signers, and sizes. Additionally, two of these datasets are recorded in an unconstrained environment, making them useful for evaluating the generalization and robustness of the proposed model.

**Experiments setup.** All experiments have been conducted on NVIDIA RTX A6000 48 GB GPU with a batch size of 16. The model was implemented using PyTorch, with a total of 58.6 million trainable parameters. The average training time was 2.7 s per sample. During inference, the model achieves an average latency of 439.03 milliseconds per sample. This inference time makes the proposed model suitable for real-time SLR. The frames were resized to $256 \times 256$ pixels, and several random augmentations, including horizontal flipping, resized cropping, and RandomAugment (*Cubuk et al., 2020*), were used. We empirically selected the number of frames of each dataset to feed the model. For WLASL-100, we extracted 16 frames from each sign video while 24 frames were extracted from other datasets. These frames were selected from the whole sign frames sequence with an

**Table 2 Information about the datasets used in this work.**

| Dataset | Sign language | No. of signs | No. of signers | No. of sample | Resolution | Environment | | Domain | |
| --- | --- | --- | --- | --- | --- | --- | --- | --- | --- |
| | | | | | | Constrained | Unconstrained | Fixed | Diverse |
| KArSL-100 | Arabic | 100 | 3 | 15,000 | $1{,}920 \times 1{,}080$ | ✓ | | | ✓ |
| KArSL-190 | Arabic | 190 | 3 | 28,500 | $1{,}920 \times 1{,}080$ | ✓ | | | ✓ |
| KArSL-502 | Arabic | 502 | 3 | 75,300 | $1{,}920 \times 1{,}080$ | ✓ | | | ✓ |
| WLASL-100 | American | 100 | 97 | 2,038 | $256 \times 256$ | ✓ | | ✗ | |
| LSA64 | Argentinian | 64 | 10 | 3,500 | $1{,}920 \times 1{,}080$ | ✓ | | | ✓ |
| AUSTL | Turkish | 226 | 43 | 38,336 | $512 \times 512$ | | ✓ | | ✓ |

equal sampling rate. The AdamW optimizer with a learning rate of 4E-4 and a weight decay of 0.05 was used to train the proposed model.

**Evaluation settings.** In this section, we present three different evaluation settings for SLR, each with varying levels of supervision. This range of settings enables us to evaluate the performance of our proposed model and compare it with the SOTA methods across sign languages that demand different levels of generalization.

- *Zero-shot setting:* The model is initially trained on a source dataset $\mathcal{D}_{\mathcal{S}}$ of the CLIP model and evaluated on the target sign language dataset $\mathcal{D}_{\mathcal{T}}$. Under this setting, the model is trained on $Y_{\mathcal{S}}$ classes of $\mathcal{D}_{\mathcal{S}}$ and evaluated on the $Y_{\mathcal{T}}$ classes belong to $\mathcal{D}_{\mathcal{T}}$, where $Y_{\mathcal{S}} \cap Y_{\mathcal{T}} = \phi$.
- *Few-shot setting:* Under this setting, the model is trained on $K$ shots (samples) of each class, $Y_{\mathcal{T}}$, of the target dataset $\mathcal{D}_{\mathcal{T}}$. In this work, we set $k$ to 1, 2, 4, and 8. For classes that do not have eight samples, we selected the maximum number of samples in that class.
- *Fully-supervised setting:* The model in this setting is trained on the entire training set of the downstream dataset $\mathcal{D}_{\mathcal{T}}$ and evaluated on the test set of that dataset.

All the experiments have been conducted in a signer-independent evaluation setting. This setup involves evaluating the model on signers not seen during training. This type of evaluation is more challenging than a signer-dependent setting where the model is trained and evaluated on samples belonging to the same signer.

**Results and discussion.** Table 3 shows the obtained results of our approach on KArSL, WALSL, LSA64, and AUST datasets. As shown in the table, we show our approach performance with different numbers of shots (training samples) starting with a zero-shot and we continued adding samples until we fine-tuned the model on the whole training data. For KArSL and LAS datasets, we report results in a signer-independent setting where one signer was fully used as a testing signer while other signers were used for training. This strategy was repeated for all datasets' signers and we report the average accuracy of all signers. For the WLASL and AUTSL datasets, we used the split provided with these datasets.

**Table 3 Accuracy results of SignVLM on KArSL, WLASL, LSA64, and AUTSL datasets.**

| Shots | KArSL-100 | | KArSL-190 | | KArSL-502 | | WLASL-100 | | LSA64 | | AUTSL | |
|---|---|---|---|---|---|---|---|---|---|---|---|---|
| | Top-1 | Top-5 | Top-1 | Top-5 | Top-1 | Top-5 | Top-1 | Top-5 | Top-1 | Top-5 | Top-1 | Top-5 |
| 0 | 0.95 | 5.2 | 0.6 | 2.5 | 0.18 | 1.1 | 1.2 | 3.9 | 1.6 | 7.6 | 0.43 | 2.0 |
| 1 | 44.0 | 77.4 | 31.4 | 63.5 | 17.5 | 43.7 | 1.9 | 17.4 | 25.1 | 67.4 | 5.6 | 16.5 |
| 2 | 71.9 | 93.9 | 48.8 | 82.6 | 38.3 | 69.2 | 11.2 | 39.2 | 68.4 | 95.6 | 22.1 | 51.3 |
| 4 | 81.9 | 97.5 | 68.5 | 91.6 | 60.2 | 86.0 | 20.2 | 53.9 | 96.5 | 99.9 | 46.0 | 79.6 |
| 8 | 85.9 | 98.3 | 76.5 | 95.3 | **61.1** | 87.3 | 55.8 | 87.6 | 98.3 | 100 | 63.6 | 89.0 |
| FSL | **89.4** | **99.1** | **79.3** | **95.9** | 60.4 | **87.9** | **79.1** | **94.2** | **99.4** | 100 | **84.6** | **97.8** |

**Note:**
Bold values are the highest score.

We started our experiments with zero-shot learning where we used the pre-trained CLIP model without fine-tuning it with our datasets. Although CLIP has outstanding performance on several image classification downstream tasks, it fails to correctly recognize sign gestures. The majority of sign language gestures are dynamic where the motion is a basic component of the sign. CLIP model was not trained on videos and it did get exposure to learn sign gestures. However, training the model with only one sample of each class with a 1-shot learning setting improved the model performance significantly by 43%, 30%, and 16.5% with KArSL-100, KArSL-300, and KArSL-502, respectively. The improvement with WLASL-100 and AUTSL datasets was not as substantial as the KArSL and LAS datasets. Adding another sample for training our model with a 2-shot learning strategy improved the performance of our approach on all datasets sharply compared with 1-shot learning. For KArSL-502, more than 20% improvement was obtained, while the model performance improved by around 10%, 43%, and 16% on WLASL-100, LSA, and AUTSL datasets, respectively, compared with 1-shot setting. As shown in Fig. 2, adding more samples significantly improved the performance of the SignVLM model on all datasets. This fast improvement demonstrates the learning capability of SignVLM in learning sign gestures when adding more samples.

To better understand the performance of the SignVLM model, we present samples of misclassified signs along with the model's predictions in Fig. 3. Some of these misclassified signs rely on fingerspelling as a key component. As shown in Fig. 3, the main difference between the FORTY and FIFTY signs in the KArSL dataset is using the thumb. A similar error is observed in the AUSTL dataset with the CRY and GOVERNMENT signs, where both signs exhibit nearly identical motion, differing only in the orientation of the hand palm. These errors highlight the SignVLM model's limitations in accurately distinguishing some signs that depend on fingerspelling.

**Comparison with other works.** Table 4 compares the performance of SignVLM against prior works on the KArSL, WLASL-100, LSA64, and AUTSL datasets. The comparison is organized based on the input modality used, pose-based or RGB-based. It is important to

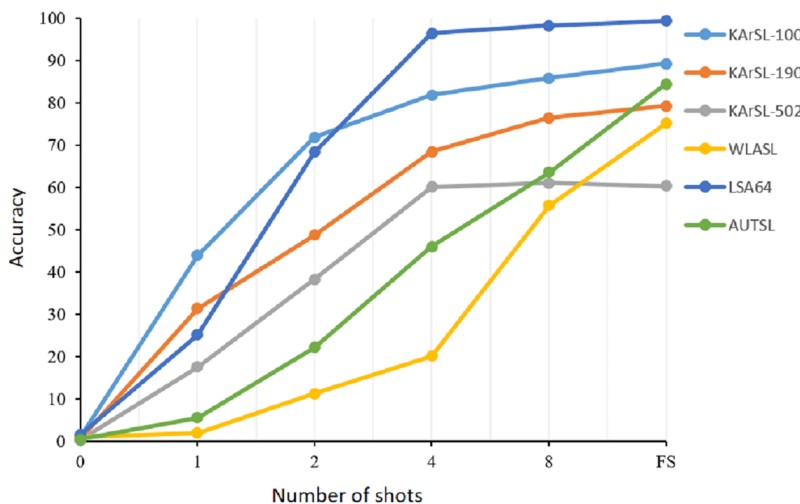

**Figure 2 The performance of the SignVLM model with a different number of shots.**

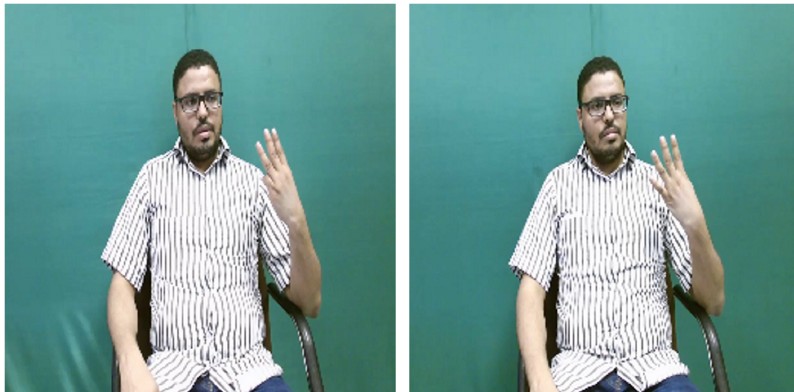

**Figure 3 A misclassified sign by SignVLM.** FORTY sign (left) and FIFTY sign (right).

note that we excluded multi-modality methods from the comparison to ensure a fair evaluation. As shown in the table, SignVLM achieves SOTA performance on the KArSL and LSA64 datasets, outperforming both pose-based and RGB-based methods. On WLASL-100, our model outperforms other RGB-based approaches and achieves competitive accuracy when compared to pose-based techniques such as SignBERT and BEST.

While our approach performs well on the AUTSL dataset, it trails the current best-performing method by approximately eight percentage points. The leading method, proposed by *Sincan & Keles (2022)*, utilizes extensive data augmentation and fuses motion-based features with 3D CNNs through a multi-stream architecture. However, this method incurs a higher computational cost due to the generation of motion history images and the parallel processing of multiple input modalities. In contrast, SignVLM offers a

**Table 4 Comparison with other works on the KArSL, WLASL, LSA64, and AUSTL datasets, based on accuracy metric.**

| Model | KArSL-100 | KArSL-190 | KArSL-502 | WLASL-100 | LSA64 | AUTSL |
|---|---|---|---|---|---|---|
| **Pose-based** | | | | | | |
| Pose-Transformer (*Alyami, Luqman & Hammoudeh, 2024*) | 68.2 | – | – | – | – | – |
| ST-GCN (*Song et al., 2017*) | – | – | – | 51.6 | – | – |
| Pose-TGCN (*Li et al., 2020b*) | – | – | – | 55.4 | – | – |
| PSLR (*Tunga, Nuthalapati & Wachs, 2021b*) | – | – | – | 60.2 | – | – |
| SignGraph (*Naz et al., 2023*) | – | – | – | 72.1 | – | – |
| SignBERT (*Hu et al., 2021*) | – | – | – | 76.4 | – | – |
| BEST (*Zhao et al., 2023a*) | – | – | – | 77.9 | – | – |
| *Marais et al. (2022)* | – | – | – | – | 81.2 | – |
| DeepSign (*Shah, 2018*) | – | – | – | – | 96.0 | – |
| *Akdag & Baykan (2024)* | – | – | – | – | 98.9 | – |
| **RGB-based** | | | | | | |
| VGG-16 (*Sidig et al., 2021*) | – | 25.6 | – | – | – | – |
| Bi-SRN (*Luqman, 2022*) | – | 40.2 | 32.7 | – | – | – |
| SCT-DCNN (*Alamri et al., 2024*) | 29.4 | 40.1 | 30.5 | – | – | – |
| I3D (*Huang et al., 2018*) | – | – | – | 67.0 | – | – |
| BoW + SubCls (*Ronchetti et al., 2016b*) | – | – | – | – | 91.7 | – |
| SD-VLAD (*Rodríguez & Martínez, 2018*) | – | – | – | – | 85.0 | – |
| Bi-SRN (*Luqman, 2022*) | – | – | – | – | 91.8 | – |
| C3D (*Li et al., 2017*) | – | – | – | – | 98.8 | – |
| BLSTM-Attention (*Sincan & Keles, 2020*) | – | – | – | – | – | 47.6 |
| S3D (*Vázquez-Enríquez et al., 2021*) | – | – | – | – | – | 90.3 |
| I3D + RGB-MHI (*Sincan & Keles, 2022*) | – | – | – | – | – | **93.5** |
| SignVLM (ours) | **89.4** | **79.3** | **60.4** | **79.1** | **99.4** | 84.6 |

**Note:**
Bold values are the highest score.

simpler and more generalizable framework. It uses a single RGB stream and a frozen visual encoder, enabling data-efficient learning and competitive performance across diverse datasets. While the performance gap on AUTSL is acknowledged, our approach demonstrates strong cross-dataset generalization and is particularly effective in low-resource scenarios, including zero-shot and few-shot learning tasks.

## ABLATION STUDY

**Number of frames.** We explored how varying the number of frames per video impacts the performance of our SLR framework across multiple datasets. Specifically, we evaluated frame counts of 8, 16, 24, and 32 with all datasets. As shown in Table 5, the obtained results highlight differences in optimal frame counts for achieving the best accuracy across these datasets. For the WLASL-100 dataset, the model achieved its highest performance with 16 frames. Increasing the frame number beyond 16 did not yield further improvements; instead, it slightly reduced performance with 24 and 32 frames. This suggests that WLASL-100 may contain signs that require less temporal information to be recognized effectively,

**Table 5 The impact of the number of frames on the performance of SignVLM.** For KArSL and LSA64, we selected signers 3 and 5, respectively.

| Frames | KArSL-190 | | WLASL-100 | | LSA64 | | AUSTL | |
|---|---|---|---|---|---|---|---|---|
| | Top-1 | Top-5 | Top-1 | Top-5 | Top-1 | Top-5 | Top-1 | Top-5 |
| 8 | 81.93 | 97.1 | 75.2 | 94.96 | 95.3 | 99.7 | 80.3 | 94.2 |
| 16 | 81.2 | 97.3 | **79.1** | 94.2 | 99.1 | 100 | 82.0 | 97.3 |
| 24 | **82.6** | 97.0 | 75.2 | 75.2 | **99.7** | 100 | **84.6** | 97.8 |
| 32 | 82.1 | 97.0 | 68.6 | 91.9 | 99.3 | 100 | 80.1 | 96.7 |

**Note:**
Bold values are the highest score.

and 16 frames capture enough context for optimal performance. In contrast, the KArSL-190, LAS64, and AUTSL datasets achieved their best results using 24 frames. For these datasets, 24 frames appear to provide the right amount of temporal information to accurately capture and interpret gestures. This variation indicates that different sign languages and datasets may have unique temporal characteristics, and some may benefit from additional context across frames.

**Learning rate.** To evaluate the impact of the learning rate on model performance, several experiments have been conducted using different learning rates. As shown in Table 6, the learning rate significantly impacts model performance. A learning rate of E-5 yielded the best results across all datasets, striking a balance between convergence and stability. High learning rates (*e.g.*, 0.001) led to suboptimal performance, likely due to overshooting, while very low rates (*e.g.*, 0.000004) caused poor convergence. The sensitivity to learning rates varied among datasets, with WLASL-100 showing the most notable differences. This highlights the importance of carefully tuning the learning rate for optimal recognition results.

**Number of decoder layers.** To evaluate the influence of decoder depth on the model's performance, we conducted an ablation study by varying the number of decoder layers in the temporal learning module of the SignVLM. Specifically, we experimented with 1, 2, 4, and 8 decoder layers. As shown in Table 7, increasing the number of decoder layers generally enhances the model's recognition accuracy across all datasets. The most substantial performance gain occurs when moving from one to four layers, which highlights the importance of deeper temporal modeling to capture complex temporal dependencies in sign sequences. However, further increasing the number of decoder layers to eight yields only marginal improvements or even slight degradations in some cases. For instance, the accuracy of the AUTSL dataset declines when increasing from four to eight layers. This degradation can be attributed to model overfitting due to model over-complexity. Based on this analysis, we conducted all our experiments using four decoder layers to balance between model complexity, generalization performance, and computational efficiency.

**Table 6 The impact of the learning rate on the performance of SignVLM.** For KArSL and LSA64, we selected signers 3 and 5, respectively.

| Learning rate | KArSL-190 | | WLASL-100 | | LSA64 | | AUSTL | |
|---|---|---|---|---|---|---|---|---|
| | Top-1 | Top-5 | Top-1 | Top-5 | Top-1 | Top-5 | Top-1 | Top-5 |
| 0.001 | 63.0 | 89.6 | 62.4 | 86.4 | 98.1 | 100 | 56.2 | 86.0 |
| 0.0001 | 82.0 | 96.9 | 72.1 | 93.4 | 99.3 | 100 | 79.4 | 96.9 |
| 0.0004 | 65.7 | 91.7 | 72.5 | 91.9 | 99.7 | 100 | 72.8 | 94.6 |
| 0.00004 | **80.1** | 95.5 | **79.1** | 94.2 | **100** | 100 | **84.6** | 97.8 |
| 0.000004 | 46.3 | 75.1 | 03.1 | 10.5 | 90.6 | 100 | 00.5 | 02.4 |

Note:
Bold values are the highest score.

**Table 7 Ablation study results with different numbers of decoder layers.**

| Number of layers | KArSL-190 | | WLASL-100 | | LSA64 | | AUSTL | |
|---|---|---|---|---|---|---|---|---|
| | Top-1 | Top-5 | Top-1 | Top-5 | Top-1 | Top-5 | Top-1 | Top-5 |
| 1 | 78.0 | 95.6 | 70.5 | 94.2 | 99.1 | 100 | 52.7 | 84.7 |
| 2 | 78.4 | 96.9 | 71.3 | 94.6 | 99.1 | 100 | 67.9 | 92.5 |
| 4 | 82.5 | 97.0 | 79.1 | 94.2 | **100** | 100 | **84.6** | 97.8 |
| 8 | **83.9** | 96.8 | **79.5** | 94.6 | 99.7 | 100 | 78.7 | 96.6 |

Note:
Bold values are the highest score.

**Table 8 The performance of SignVLM when varying the number of decoder attention heads.**

| Number of heads | KArSL-190 | | WLASL-100 | | LSA64 | | AUSTL | |
|---|---|---|---|---|---|---|---|---|
| | Top-1 | Top-5 | Top-1 | Top-5 | Top-1 | Top-5 | Top-1 | Top-5 |
| 2 | 75.1 | 94.4 | 72.9 | 92.6 | 99.7 | 100 | 73.7 | 95.4 |
| 4 | 74.6 | 94.4 | 74.4 | 93.4 | 99.7 | 100 | 75.1 | 95.2 |
| 8 | 82.4 | 97.9 | 74.8 | 92.3 | 99.7 | 100 | 72.4 | 94.0 |
| 16 | **82.5** | 97.0 | **79.1** | 94.2 | **100** | 100 | **84.6** | 97.8 |
| 32 | 82.2 | 97.2 | 69.8 | 95.0 | 99.7 | 100 | 66.7 | 91.3 |

Note:
Bold values are the highest score.

**Number of decoder heads.** Table 8 shows the impact of varying the number of decoder attention heads on the performance of the SignVLM model. The results demonstrate that increasing the number of heads generally enhances performance, particularly when moving from a shallow attention configuration (2 or 4 heads) to a moderately wider setup. The most notable improvements occur at 8 and 16 heads, where the model achieves strong accuracy gains across all datasets. For instance, the Top-1 accuracy on KArSL-190 increases from 74.6% (4 heads) to 82.5% (16 heads), while WLASL-100 peaks at 79.1% with 16 heads. Similarly, on the AUSTL dataset, the model achieves its highest Top-1 accuracy of 84.6% with 16 heads, indicating that a broader multi-head attention setup helps the model capture diverse temporal patterns and signer variability more effectively.

However, performance slightly degrades when increasing to 32 heads, particularly on datasets like AUTSL and WLASL-100, where the Top-1 accuracy drops to 66.7% and 69.8%, respectively. This decline suggests that large numbers of attention heads may lead to model overfitting, noise sensitivity, or reduced attention effectiveness due to the diluted focus of each head. Based on these findings, we selected 16 decoder heads as the optimal configuration, which provides the best trade-off between accuracy and computational efficiency across varied sign language datasets

## CONCLUSIONS

This work explored the adaptation of a large vision-language model, CLIP, for isolated SLR. We proposed SignVLM, a simple yet effective framework that integrates a frozen CLIP visual encoder with a Transformer-based temporal decoder. Unlike many prior works, our approach emphasizes data efficiency, which enables effective learning under zero-shot and few-shot conditions.

The proposed approach has been evaluated on four sign language datasets KArSL, WLASL-100, LSA64, and AUTSL. These datasets represent different sign languages and resource levels. The model achieved SOTA performance on KArSL and LSA64, outperformed RGB-based approaches on WLASL-100, and remained competitive on AUTSL, despite using a single RGB modality and a lightweight architecture. These results demonstrate that CLIP's spatial embeddings, when combined with temporal modeling, can generalize effectively to some sign languages without requiring extensive dataset-specific tuning. In addition, the proposed model shows practical advantages in low-resource settings, offering a scalable direction for future SLR systems, especially for underrepresented languages. Future work may explore extending this framework to continuous SLR and integrating language-aware decoders to enhance semantic interpretation beyond isolated gloss recognition.

### Funding
This work was supported by the Saudi Data and AI Authority (SDAIA) and King Fahd University of Petroleum and Minerals (KFUPM) under the SDAIA-KFUPM Joint Research Center for Artificial Intelligence Grant No. JRC-AI-RFP-14. The funders had no role in study design, data collection and analysis, decision to publish, or preparation of the manuscript.

### Grant Disclosures
The following grant information was disclosed by the authors:
Saudi Data and AI Authority (SDAIA) and King Fahd University of Petroleum and Minerals (KFUPM): JRC-AI-RFP-14.

## Competing Interests

The authors declare that they have no competing interests.

## Author Contributions

- Hamzah Luqman conceived and designed the experiments, performed the experiments, analyzed the data, performed the computation work, prepared figures and/or tables, authored or reviewed drafts of the article, and approved the final draft.

## Data Availability

Code is available at GitHub:

https://github.com/Hamzah-Luqman/signVLM

https://github.com/OpenGVLab/efficient-video-recognition?tab=readme-ov-file

The WLASL dataset is available at: https://dxli94.github.io/WLASL/

The KArSL: dataset is available at: https://hamzah-luqman.github.io/KArSL/

The LSA64 dataset is available at: https://facundoq.github.io/datasets/lsa64/

The AUTSL dataset is available at: https://cvml.ankara.edu.tr/datasets/

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
