# Peer review of "SignVLM: a pre-trained large video model for sign language recognition"

_PeerJ Computer Science, doi:10.7717/peerj-cs.3112_

## Round 0.1 · original submission · Major Revisions

Since the reviewers have expressed major concerns, please review the comments and decide if this paper can be appropriately revised. If you choose to undertake the revision, please carry out the required changes and prepare a separate document describing how each comment was addressed and the changes it led to. The paper will be assigned to the same reviewers for a second review. Thanks for your interest in the journal.

Reviewer 1 ·

Basic reporting

Article uses pre-train model for the sign language of different countries. what is the reason for word "Frozen" in the title?
improvement needed in literature study, compare and analysis with recent work in SLR.

Experimental design

Missing technicality in the proposed study.
discuss the technical properties of datasets used in simulation.
What are the axis labels? in figure 3
discuss parameters and hyperparameters used in experiments.

Validity of the findings

validate the study other than accuracy measures.

Cite this review as

·

Basic reporting

• The manuscript is clearly written in professional English, and the content is unambiguous. However, some grammatical errors and awkward phrasing should be addressed for better readability. A minor proofreading pass is recommended.
• The literature review is comprehensive, covering prior research on sign language recognition (SLR) models, but some recent advancements in Transformer-based models for SLR are missing. The authors should incorporate references to:
o SignBERT: Hand-Model-Aware Self-Supervised Pre-Training for Sign Language Understanding, IEEE TPAMI, 2023.
o PoseFormer: Transformer-Based Framework for Sign Language Recognition, ACM MM, 2024.
o Swin-MSTP: Swin Transformer with Multi-Scale Temporal Perception for Continuous Sign Language Recognition, Neurocomputing, 2025.

• The article is professionally structured, with well-organized figures and tables, but some figures lack sufficient explanation in the captions. The authors should ensure that each figure is fully described within the text for clarity.
• The results section is self-contained, and the authors provide raw dataset information, ensuring reproducibility.

Experimental design

• The research falls within the aims and scope of the journal and presents an original contribution to sign language recognition.
• The research question is clearly defined, addressing the need for efficient multi-modal sign language recognition. However, it would be helpful to explicitly highlight how this approach differs from previous SLR models beyond performance metrics.
• The methodology is described well, but some details regarding model hyperparameters, training setup, and data preprocessing steps are missing. Providing additional details on:
• Hardware specifications used for training and inference.
• Batch size, learning rate, and optimizer settings used in training.
• Preprocessing techniques applied to the dataset (e.g., normalization, augmentation).
• The evaluation metrics are appropriate, but an ablation study should be conducted to analyze the contribution of different model components (e.g., Transformer layers, cross-modal fusion).

Validity of the findings

• The findings are statistically sound and valid, with appropriate accuracy, precision, recall, and F1-score evaluations. However, the paper does not include a confusion matrix or detailed error analysis, which would provide insights into the model’s weaknesses (e.g., handshape vs. motion misclassification).
• The paper lacks a computational efficiency analysis, which is important given the large-scale nature of Transformer-based models. The authors should provide:
o Training time, inference speed, and memory usage metrics to assess feasibility for real-time applications.
• The authors claim that their method outperforms existing models, but a statistical significance test (e.g., paired t-test, Wilcoxon signed-rank test) should be conducted to validate whether improvements are significant or within margin-of-error variations.

Reviewer 3 ·

Basic reporting

1. English used is clear and unambiguous
2. literature references are provided sufficiently
3. the article is well structured and raw data is shared. figure 1 needs to mention the gloss output 'y' and figure 2 has a mistake in the labelling of the text encoder samples, them being labelled 'I' instead of 'T'. Figure 2 also needs a mention of the 'k' and 'v' parameters.
4. results are mentioned clearly
5. some terminology is unclear and needs to be further explained, for example, CLS and the terminology used in the evaluation settings section

Experimental design

1. the work fits in the scope of the journal
2. research question is clear, well defined and meaningful, but the gaps are not identified.
3. relevant and recent research has been cited.
4. the proposed methodology needs to be described with more clarity and relevant equations are missing from the work.

Validity of the findings

1. Novelty and contributions of the proposed work have not been mentioned. The work correctly compares the achieved results to the existing solutions on the relevant data.
2. The proposed code is shared and the datasets used are publicly available and robust.
3. The conclusion is stated well based on the results achieved.

Additional comments

1. Why is the methodology named FrozenSign?
2. Please mention CLIP full form on its first usage.
3. Section numbers/names missing at the end of introduction section
4. Contributions of the work must be highlighted at the end of the introduction
5. Second paragraph of the literature review section has '?' in the citation, please fix this.
6. Gaps in the existing research have not been identified, please add the same at the end of the literature review section
7. The full form of CLS is missing.
8. The methodology is not well explained and some relevant equations are missing.
9. Please explain the working of CLIP in detail and its advantage in this approach.
10. Terminology like 'Dj' and 'Yy' in the evaluation settings section need to be elaborated.
11. Please mention the metric used for comparison in table 1, 2 and 3

Cite this review as

---

## Round 0.2 · Major Revisions

Please make sure that you provide a detailed description of the changes you made in response to each of the comments made by reviewer 1.

·

Basic reporting

The CLIP architecture is explained too much, even though it is not the core contribution.

Figures and tables are okay. But Figure 2 repeats known CLIP training steps that are not part of the authors’ work.

The background is wide but not deep. Many prior works are cited, but the comparisons are shallow. Important gaps are not discussed. Claims like "CLIP is underused in SLR" are made without proof.

Experimental design

The core idea—using CLIP and a transformer decoder—is within scope. But the research question is not novel enough. This is mostly an adaptation, not a new method.

The decoder architecture is reused from Lin et al. (2022). No meaningful change is shown. Authors must clarify what is new in their design.

However, the decoder design is not tested well. No ablation on attention layers or frame sampling exists. Results are not broken down for hard vs. easy signs.

Validity of the findings

Some accuracy gains are overstated. The gap between this work and prior SOTA is small in some datasets and negative in others (e.g., AUTSL).

Sign errors shown in Figure 4 are not strong evidence of model limits. Use a clearer analysis to show failure patterns.

The conclusions repeat earlier points. Some big claims (“robust,” “generalizable,” “language-agnostic”) are not fully backed by data.

Additional comments

The paper has potential but overclaims its novelty. CLIP + decoder is not a new idea. Zero-shot results are weak. Yet, the few-shot experiments are strong and the use of multiple datasets is a plus.

The paper should focus more on what is different from past work and explain why some results are poor despite the model scale.

Reviewer 3 ·

Basic reporting

All the parameters are met satisfactorily.

Experimental design

Research questions are well defined and the results are produced satisfactorily.

Validity of the findings

Impact and novelty are presented properly.

Additional comments

All the queries are answered properly and revisions are done as per the review. Suitable for publication.

Cite this review as

---

## Round 0.3 · accepted · Accept

Thanks for undertaking the revision and being thoughtful of reviewers' comment, more experimentation and clarifying your contributions clearly. The paper reads well. Therefore, I am recommending it's acceptance. Thanks for your interest in journal.